# Novel Freshwater Ascomycetes from Spain

**DOI:** 10.3390/jof8080849

**Published:** 2022-08-14

**Authors:** Viridiana Magaña-Dueñas, José Francisco Cano-Lira, Alberto Miguel Stchigel

**Affiliations:** Mycology Unit, Medical School, Universitat Rovira i Virgili, C/Sant Llorenç 21, 43201 Reus, Spain

**Keywords:** ascomycetes, *Dothideomycete*, freshwater, fungi, *Leotiomycetes*, taxonomy

## Abstract

Freshwater ascomycetes are a group of fungi of great ecological importance because they are involved in decomposition processes and the recycling of organic matter in aquatic ecosystems. The taxonomy of these fungi is complex, with representatives in several orders of the phylum Ascomycota. In the present study, we collected ninety-two samples of plant debris submerged in freshwater in different locations in Spain. The plant specimens were placed in wet chambers and developed several fungi that were later isolated in pure culture. A main phylogenetic tree using the nucleotide sequences of D1–D2 domains of the 28S nrRNA gene (LSU) was built to show the taxonomic placement of all our fungal strains, and, later, individual phylogenies for the different families were built using single or concatenated nucleotide sequences of the most suitable molecular markers. As a result, we found a new species of *Amniculicola* that produces a coelomycetous asexual state, a new species of *Elongatopedicellata* that produces an asexual state, a new species of *Neovaginatispora* that forms both sexual and asexual states in vitro, and the sexual states of two species of *Pyrenochaetopsis,* none of which have been reported before for these genera. In addition, we describe a new species of *Pilidium* characterized by the production of copper-colored globose conidiomata, and of *Pseudosigmoidea*, which produces well-developed conidiophores.

## 1. Introduction

The phylum Ascomycota is a monophyletic group of fungi that includes several taxa, commonly from freshwater habitats where they complete a part or the whole of their life cycle [1,2]. They decompose substrates such as stems, rotten wood, and decaying leaves, which then fall into bodies of water adjacent to vegetation [3]. These fungi produce a rich array of enzymes that degrade cellulose, hemicellulose, and lignin, among other natural polymers present in plant tissues. They not only provide assimilable nutrients for themselves, but also for other organisms in the same ecological niches [4]. Freshwater habitats are characterized by having a balance of organic matter controlled by the surface, location, and characteristics of the watershed [5] and can be described as lotic or lentic. Lotic environments are aquatic ecosystems with a constant flow of water (rivers, streams, springs, etc.). Lentic environments, by contrast, lack a constant flow of water (lakes, ponds, swamps, etc.) [6].

Aquatic fungi are classified into various ecological groups according to their level of adaptation and activity, and their dependence on aquatic environments [7]. Resident or native fungi are those fully adapted to an aquatic lifestyle and show morphological and physiological adaptations; most of them are also capable of sporulating below the water surface [7]. Periodic immigrant or amphibian fungi inhabit aquatic environments during parts of their life cycle and have the ability to produce spores on the land and in the water [7,8]. Versatile or facultative aquatic immigrant fungi are poorly adapted to water environments and do not sporulate below the water surface. Finally, transient fungi are those not at all adapted to aquatic environments, and are unable to sporulate or, consequently, colonize a new substrate [7].

In recent years, knowledge of the taxonomy and phylogeny of these fungi has been increasing thanks to the use of molecular techniques that mostly use PCR-based amplification and sequencing of different sorts of genes [9,10,11]. To date, ca. 3000 species of ascomycetes have been described from freshwater habitats [12]. Freshwater ascomycetes are represented in several orders and families scattered across four main classes: the Dothideomycetes (677 species), the Eurotiomycetes (276 species), the Leotiomycetes (260 species), and the Sordariomycetes (823 species) [12].

The hydrography in Spain is peculiar and inconsistent due to the richness and diversity of waterways, ecosystems, and landscapes (https://hispagua.cedex.es/en/datos/hidrografia#1; accessed on 3 April 2022). There are four drainage areas in Spain: North, South, East, and West. Apart from the large northern rivers, the Duero and the Ebro, many have very low water flow because of low rainfall. Their profiles vary from the fast rivers in the Cantabrian slope and the Pyrenees to the slow rivers of the central plateau. Except for the Ebro, the rivers of the Atlantic slope are the longest [13]. Spain is a country where the hyphomycetes have been studied for decades, but only in recent years have the sexual states of freshwater ascomycetes and their coelomycetous asexual states begun to be investigated with any great frequency [14,15,16]. The present study describes the phenotype and taxonomy of several new species of the Ascomycota found in various freshwater habitats in Spain, all of them isolated from submerged plant debris. Their characterization and placement were achievable through the sequencing of several phylogenetically informative markers.

## 2. Materials and Methods

### 2.1. Samples Collection and Fungal Isolation

Ninety-two samples of plant material, consisting of small branches, leaves, and bark, were collected from different lotic environments in Spain (Figure 1): 50 from *Cascadas del Huéznar* (*Cazalla de la Sierra*, *Sevilla* province, Spain); 17 from the *Riaza* river (near *Riaza*, *Segovia* province, Spain); 22 from *de les Hortes* river (on the outskirts of *Capafonts*, *Tarragona* province, Spain); three from *Clot de la Mare de Déu* (Burriana, *Castellón* province, Spain).

*Cascadas del Huéznar* (37.993824, −5.668985) forms a part of the course of the *Rivera del Huéznar* river in the *Sierra Norte de Sevilla* natural park. The sampling location was at an altitude of around 600 m.a.s.l. It has a Csa climate (hot-summer Mediterranean climate) according to the Köppen–Geiger climate classification [17], the average annual temperature is 16 °C, and the average annual rainfall is 540 mm (https://es.climate-data.org/europe/espana/andalucia/san-nicolas-del-puerto-828536/; accessed on 3 April 2022). Impressive travertine formations stand out in this waterfall and there is an abundance of riverside vegetation that forms a dense gallery forest of alders, ash trees, elms, and willows (https://www.juntadeandalucia.es/medioambiente/portal/web/ventanadelvisitante/detalle-buscador-mapa/-/asset_publisher/Jlbxh2qB3NwR/content/cascadas-del-huesna-2/255035; accessed on 3 April 2022).

The *Riaza* river (41.28368, −3.47187) also has a riverside forest with an abundance of alders, ash trees, elms, and poplars. The sampling location was at around 1190 m.a.s.l. It has a Cfb climate (temperate oceanic climate without a dry season), an average annual temperature of 10.8 °C, and a total annual rainfall of around 690 mm (https://es.climate-data.org/europe/espana/castilla-y-leon/riaza-188771/; accessed on 3 April 2022). Soils are acidic, with a pH of around 5, conditioned by the presence of siliceous materials [18].

The *de les Hortes* river (41.28664, 1.04033) runs on a calcareous soil and is surrounded by a forest of boxwoods of considerable dimensions, holm oaks, pines, and poplars, among other trees and shrubs (http://www.valldecapafonts.com/capafonts_racons_llodriga.html; accessed on 3 April 2022). The sampling location has a Csa climate, an average annual temperature of 13 °C, and an average annual rainfall of 525 mm (https://es.climate-data.org/europe/espana/cataluna/capafonts-662465/; accessed on 3 April 2022).

The *Clot de la Mare de Déu* municipal natural area (39.88043, −0.05954) originates from springs in the last part of the *Sec river* (or *Anna river*), which passes through the city of *Burriana*. It is a representative example of the riparian forest, composed of elms, hackberries, poplars, and willows, together with an abundance of shrubs, herbaceous and aquatic vegetation of reeds and rushes (https://turisme.burriana.es/listing/el-clot-de-la-mare-de-deu/; accessed on 3 April 2022). The soil is made up of ochre-colored clays and silts (https://www.burriana.es/ayuninf/tablon/ORDENACION/EVALUACION-AMBIENTAL-PGOU/ESTUDIO-DE-PAISAJE/MEMORIA_PDF/MEMORIA.pdf; accessed on 3 April 2022). The local climate is BSk (cold semi-arid), with an average annual temperature of around 17 °C, and average annual precipitation of 400 mm (https://es.climate-data.org/europe/espana/comunidad-valenciana/burriana-56924/; accessed on 3 April 2022).

Samples were placed into self-sealing sterile plastic bags, closed, and transported to the laboratory where they were stored at room temperature (20–25 °C) for four–seven days. The plant debris was rinsed twice with 500 mL sterile tap water, placed into 15 cm diameter disposable Petri dishes lined inside with two layers of filter paper moistened with sterile water (1 mL of a solution of 20 mg Dieldrin^®^ in 20 mL of dimethyl-ketone/L of water), incubated at room temperature (20–25 °C), and examined periodically with a stereo microscope for between four weeks and two months, until reproductive (asexual or/and sexual) structure development. A number of single fungal propagules (ascospores or conidia) were transferred using sterile disposable needles to 55 mm diameter disposable Petri dishes containing oatmeal agar (OA; 30 g of filtered oat flakes, 15 g agar-agar, 1 L tap water; [19], and then incubated at 25 ± 1 °C. This was repeated until a pure culture was obtained, and the strains of interest were deposited in the culture collection at the Faculty of Medicine (FMR, Reus, Spain) in three different ways: slant cultures on OA and potato dextrose agar (PDA; Pronadisa, Madrid, Spain) under a layer of liquid vaseline; OA blocks (where the strain had grown) immersed in sterile water in caramel-colored self-sealing vials; and lyophilized. Holotypes and cultures ex-type of the novel fungal taxa were deposited in the Westerdijk Fungal Biodiversity Institute (CBS; Utrecht, The Netherlands). The names and descriptions were deposited in MycoBank.

### 2.2. Phenotypic Study

Fungal strains were characterized macroscopically following incubation on OA, PDA, and MEA (malt extract agar; 40 g of malt extract, 15 g of agar-agar, 1 L distilled water) at 25 + 1 °C for 14 days, annotating the colony diameter and reporting the texture, topography, margins, presence of diffusible pigments and exudates. Colony color (surface and reverse) was described according to Kornerup and Wanscher [20]. The cardinal temperatures of growth were determined on PDA after 7 days of incubation in darkness, ranging from 5 to 35 °C, at 5 °C intervals. For each fungal structure of taxonomic interest, the measurements of ten specimens were carried out on Shear’s mounting medium (3 g potassium acetate, 60 mL glycerol, 90 mL ethanol 95%, and 150 mL distilled water [21]). Material was taken from the natural substrate and/or from the fungal strains grown on OA in the same conditions as for colony characterization. Histological sections of the sexual or asexual reproductive bodies were made freehand with the help of a sterile 0.3 × 13 mm needle and a 15 mm wide Laseredge Scalpel. Photomicrographs were taken using a Zeiss Axio-Imager M1 microscope (Oberkochen, Germany) with a DeltaPix Infinity X digital camera using Nomarski differential interference contrast.

### 2.3. DNA Extraction, Amplification, and Sequencing

To extract genomic DNA, the mycelium of axenic cultures that had been grown in PDA for 7 days at 25 ± 1 °C in the dark was scraped with a sterile scalpel. The DNA was then extracted using the FastaDNA kit protocol (Bio101, Vista, CA, USA) with a Fast Prep FP120 instrument (Thermo Savant, Holbrook, NY, USA) according to the manufacturer’s protocol. DNA was quantified using Nanodrop 2000 (Thermo Scientific, Madrid, Spain). The following loci were amplified and sequenced: internal transcribed spacer region (ITS), with the primer pair ITS5 and ITS4 [22]; a fragment of the 28S nrRNA gene (LSU) with the primer pair LR0R [23] and LR5 [24]; a fragment of the RNA polymerase II subunit 2 gene (*rpb2*) with RPB2-5F2 [25] and fRPB2-7cR [26]; a fragment of the beta-tubulin gene (*tub2*) with the primers TUB2Fw and TUB4Rd [27]; and translation elongation factor 1-alpha (*tef1*) with the primers EF1-983F and EF1-2218R [28]. Amplicons were sequenced in both directions with the same primer pair used for amplification at Macrogen Spain (Macrogen Inc., Madrid, Spain). The sequences obtained were edited and contigs were assembled using SeqMan software v. 7.0.0 (DNAStar Lasergene, Madison, WI, USA). Sequences generated in this study were deposited in GenBank (https://www.ncbi.nlm.nih.gov/genbank/; accessed on 7 Juny 2022) (Appendix A).

### 2.4. Phylogenetic Analysis

Each sequence generated in this study was subjected to an individual BLAST search to verify its identity in the National Center for Biotechnology Information (NCBI) database using the Basic Local Alignment Search Tool (BLAST; https://blast.ncbi.nlm.nih.gov/Blast.cgi; accessed on 11 October 2021). Fungal strains were identified at species level when the nucleotide sequences of selected loci displayed a level of identity ≥ 98% with those of the ex-type or reference strains in the database. The loci used for this purpose were: ITS for *Pilidium* spp. and *Pseudosigmoidea* spp.; LSU for *Elongatopendicellata* spp.; *rpb2* for *Neovaginatispora* spp. and *Pyrenochaetopsis* spp.; and *tef-1* for *Amniculicola* spp. Each locus was aligned with the MEGA (Molecular Evolutionary Genetics Analysis) software v. 7.0 [29], using the ClustalW algorithm [30] and refined with MUSCLE [31] or manually, if necessary, using the same software. Phylogeny was analyzed by maximum-likelihood (ML) and Bayesian inference (BI) with RAxML v. 8.2.12 [32] software on the online Cipres Science gateway portal [33] and MrBayes v.3.2.6 [34], respectively.

A main LSU phylogenetic tree was built to display the taxonomic placement of all our fungal strains at family and genus level, using nucleotide sequences of representatives of the families *Amniculicolaceae, Cucurbitariaceae, Leptosphaeriaceae, Lophiostomataceae, Neopyrenochaetaceae, Pleosporaceae, Pseudopyrenochaetaceae, Pyrenochaetopsidaceae, Roussoellaceae, Sympoventuriaceae, Torulaceae* and *Venturiaceae* (class *Dothideomycetes*), and *Chaetomellaceae* (class *Leotiomycetes*), including those sequences of our strains.

The phylogenetic tree for the family *Amniculicolaceae* was built using the concatenated nucleotide sequences of the ITS region and a fragment of the *tef1* gene. For the family *Roussoellaceae,* only the LSU was employed. For the family *Lophiostomataceae,* three concatenated markers—ITS, *rpb2,* and *tef1*—were used. For members of the families *Chaetomellaceae* and *Sympoventuriaceae,* the concatenated markers employed were LSU and ITS; and for the phylogenetic analysis of the family *Pyrenochaetopsidaceae,* the concatenated sequences of ITS, *rpb2,* and *tub2* were employed.

The best nucleotide substitution model for BI analysis was estimated using the jModelTest program [35]. The best model used for the main LSU phylogenetic tree was the symmetrical model with proportion of invariable sites and gamma distribution (SYM + I+G). The best model used for the *Amniculicolaceae* was SYM + I+G for ITS and the general time reversible with gamma distribution (GTR + G) for LSU and *tef1*. The best model used for the *Roussoellaceae* was the general time reversible with proportion of invariable sites and gamma distribution (GTR + I+G) for LSU. For the *Lophiostomataceae*, the best model was Kimura 2-parameter with proportion of invariable sites and gamma distribution (K80 + I+G) for ITS and *rpb2*; and GTR + G for *tef1*. The best model substitution for the *Chaetomellaceae* was SYM + I+G for ITS and K80 + I+G for LSU. For the *Sympoventuriaceae* it was K80 + I for ITS and LSU. For the *Pyrenochaetopsidaceae*, the Kimura 2-parameter with proportion of invariable sites (K80 + I) was used for ITS, the SYM + I+G for *rpb2*, and the Hasegawa–Kishino–Yano with Gamma distribution (HKY + G) for *tub2*.

The parameters used in Bayesian analysis were two simultaneous runs of 5,000,000 generations. The 50% majority rule consensus tree and posterior probability values (PP) were calculated after discarding the first 25% of the resulting trees. A PP value ≥ 0.95 was considered significant [36]. For ML analysis, support for internal branches was assessed by 1000 ML bootstrapped pseudoreplicates. Bootstrap support value (BS) ≥ 70% was considered significant. Alignments and trees were deposited in TreeBASE (http://purl.org/phylo/treebase/phylows/study/TB2:S29604; accessed on 7 Juny 2022). The novel nomenclature and novel taxonomic descriptions were deposited in MycoBank (http://www.mycobank.org; accessed on 7 Juny 2022).

## 3. Results

### 3.1. Phylogeny

The LSU phylogenetic tree for the members of the *Dothideomycetes* and the *Leotiomycetes* classes comprised 91 ingroups of species with a total of 776 characters including gaps, from which 266 bp were parsimony-informative (Figure 2). The ML analysis was congruent with that obtained by BI analysis, displaying trees with similar topologies. For the BI analysis, a total of 1574 trees were sampled after the burn-in with a stop value of 0.01. The phylogenetic tree (Figure 2) shows two main clades, the first corresponding to the *Dothideomycetes* class (100% BS/1 PP), in which six of our strains were located, and the second to the *Leotiomycetes* class (90% BS/1 PP). Two of our strains (FMR 18801 and FMR 18913) were located in the clade corresponding to the genus *Pyrenochaetopsis* (100% BS/1 PP), close to the species *P. confluens* and *P. decipiens*. FMR 17834 was located within the *Roussoellaceae* family (90% BS/1 PP), together with *Elongatopedicellata lignicola*, the type species of the genus. Within the clade of the genus *Neovaginatispora* (100% BS/1 PP), the FMR 18914 strain closely related phylogenetically to *N. clematidis* and *N. fuckelii* was located. FMR 17416 was located within the *Sympoventuriaceae* family (100% BS/1 PP), in the terminal branch (84% BS/0.97 PP), together with *Pseudosigmoidea* spp. In the respective clade of the *Amniculicolaceae* family (100% BS/1 PP), the FMR 17946 strain was located, together with species of the *Amniculicola, Fouskomenomyces,* and *Vargamyces* genera. FMR 17839 was located in *Chaetomellaceae* (90% BS/1 PP), in the terminal branch (unsupported), together with species of the genus *Pilidium*. The first main clade was divided into 12 subclades with respect to the families *Leptosphaeriaceae* (100% BS/1 PP), *Pleosporaceae* (89% BS/1 PP), *Pseudopyrenochaetaceae* (93% BS/1 PP), *Neopyrenochaetaceae* (unsupported), *Pyrenochaetopsidaceae* (100% BS/1 PP), *Cucurbitariaceae* (unsupported), *Rousoellaceae* (91% BS/1 PP), *Torulaceae* (100% BS/1 PP), *Lophiostomataceae* (98% BS/1 PP), *Sympoventuriaceae* (100% BS/1 PP), *Venturiaceae* (without support), and *Amniculicolaceae* (79% BS/0.99 PP). The second main clade was only formed by members of the *Chaetomellaceae* family (90% BS/1 PP), in which the strain FMR 17839 was located.

To clarify the phylogenetic relationship of our isolate FMR 17946 with the species of Amniculicola, a separate phylogenetic analysis was carried out. The final concatenated nucleotide sequences comprised 20 ingroups of species with a total of 1117 characters including gaps, from which 187 bp were parsimony-informative (113 for ITS and 74 for tef1) (Figure 3). The BI analysis showed a similar tree topology and was congruent with that obtained in the ML analysis. For the BI multi-locus analysis, a total of 1141 trees were sampled after the burn-in with a stop value of 0.01. In the phylogenetic tree, the Amniculicolaceae formed a full-supported clade (100% BS/1 PP) including all genera accepted in the family: Amniculicola, Fouskomenomyces, Murispora, and Vargamyces. Our strain FMR 17946 was placed in the same terminal clade as *A. guttulata*.

The phylogenetic analysis of the members of the Roussoellaceae included partial LSU nucleotide sequences from 18 species with a total of 845 characters including gaps, from which 80 bp were parsimony-informative. The topology of the trees inferred by the two phylogenetic methods (ML and BI) were basically the same, with minor differences in statistically supported groupings. For the BI analysis, a total of 385 trees were sampled after the burn-stop value of 0.01. In the phylogenetic tree (Figure 4), our strain FMR 17834 was placed in the same fully-supported terminal clade as the type species of the genus Elongatopedicellata (*E. lignicola*).

For the *Lophiostomataceae*, the combined alignment of the three loci datasets (ITS-*rpb2* and *tub2*) encompassed 15 ingroups of species with a total of 2265 characters including gaps, of which 609 bp were parsimony-informative sites (178 for ITS, 298 for *rpb2*, 129 for tub2). Phylogenies obtained by ML and BI showed a topological congruence. For the BI multi-locus analysis, a total of 402 trees were sampled after the burn-stop value of 0.01. In the *Lophiostomataceae* (Figure 5), the genus *Neovaginatispora* (100% BS/1 PP) included the two previously accepted species of the genus plus our strain FMR 18914.

We carried out a combined phylogenetic analysis with ITS and LSU sequences to resolve the taxonomic position of our strain FMR 17839 in Chaetomellaceae. A concatenated dataset from 18 sequences contained a total of 1246 characters, from which 277 bp were parsimony-informative sites (172 for ITS and 105 for LSU). Bayesian inference and ML analyses of the concatenated dataset yielded similar topologies. For the BI multilocus analysis, a total of 134 trees were sampled after the burn-stop value of 0.01. In that phylogenetic tree (Figure 6), the genus Pilidium (100%BS/1PP) included all species with sequences available as well as our strain FMR 17839, which was placed alone in a basal terminal branch.

The phylogenetic analysis of the Sympoventuriaceae included sequences from eight species, with 1325 characters including gaps, from which 161 bp were parsimony-informative sites (93 for ITS and 68 for LSU). The topology of the tree identified by ML analysis was almost identical to that found by the Bayesian analyses. For the BI multi-locus analysis, a total of 29 trees were sampled after the burn-stop value of 0.01. In our phylogenetic tree (Figure 7), the genus Pseudosigmoidea formed a well-supported clade (99% BS/1 PP) and included all accepted species (except the type species P. cranei) plus our strain FMR 17416.

The tree for the Pyrenochaetopsidaceae was built using the concatenated sequences from 24 species, totaling 1488 characters including gaps, from which 471 bp were parsimony-informative (82 for ITS, 281 for rpb2, and 107 for tub2). The topologies estimated by ML are almost the same and correspond to those obtained by the BI method. A total of 1702 trees were sampled after the burn-stop value of 0.01. In the phylogenetic tree (Figure 8), the genus Pyrenochaetopsis formed a fully supported clade including all previously accepted species and our strains FMR 18801 and FMR 18913, which were placed in two different and fully-supported terminal branches, FMR 18913 together with *P. globosa* and *P. uberiformis*, and FMR 18801 jointly with *P. confluens*.

### 3.2. Taxonomy

Amniculicolaceae Y. Zhang ter, C.L. Schoch, J. Fourn., Crous, and K.D. Hyde, Studies in Mycology 64: 95 (2009). MycoBank MB515469.
Type genus: *Amniculicola* Y. Zhang ter and K.D. Hyde, Mycological Research 112 (10): 1189 (2008). MycoBank MB 511328.Type species: *Amniculicola lignicola* Y. Zhang ter and K.D. Hyde, Mycological Research 112 (10): 1189 (2008). MycoBank MB 511330.Because the asexual state of the genus *Amniculicola* has not been reported and described yet, we have emended the generic description.*Amniculicola* Y. Zhang and K.D. Hyde, emended by V. Magaña-Dueñas, Cano, and Stchigel.Saprobic on wood in freshwater habitats. Sexual state: Ascomata solitary to gregarious, immersed to nearly superficial, black, uniloculate, subglobose to conical, glabrous, ostiolate, with or without two tuberculate fared lips surrounding a slit-like ostiole, sometimes with a fattened base not easily removed from the substrate, usually staining the woody substrate in purple tinge. Peridium 2-layered, outer layer composed of heavily pigmented thick-walled cells of *textura angularis*, inner layer composed of hyaline thin-walled cells of textura angularis. Pseudoparaphyses dense, trabeculate, filiform, persistent, hyaline, embedded in mucilage, anastomosing between and above the asci. Asci 8-spored, bitunicate, fissitunicate, cylindrical to narrowly fusoid, short pedicellate. Ascospores mostly uniseriate, fusoid, hyaline, septate, symmetrical, smooth, thin-walled, surrounded by a hyaline, gelatinous sheath. Asexual states: anguillospora-like. Conidiophores usually simple, conidia septate, hyaline, curved or sigmoid, tapering to the apex; phoma-like. Conidiomata pycnidia, dark brown, semi-immersed, solitary, scattered, pycnidial wall of *textura angularis*, globose to subglobose. Conidiogenous cells phialidic, determinate, hyaline, globose to ampulliform, flask-shaped, or cylindrical. Conidia aseptate, hyaline, smooth, thin-walled, globose to kidney-shaped or bacilliform.

*Amniculicola asexualis* V. Magaña-Dueñas, Stchigel, and Cano, sp. nov. MycoBank MB842769 (Figure 9).
Etymology: From Latin *asexualis*, without sex, because of lack of a known sexual state.Type: Spain, Sevilla province, Parque Natural Sierra Norte, Cascadas del Huéznar, from plant debris in freshwater, May 2019, José F. Cano Lira and Juan R. García Martínez, holotype CBS H-24925, living cultures FMR 17946, CBS 148919.Description: Sexual state: unknown. Asexual state (found in natural substrate but described in vitro): Hyphae hyaline to pale brown, septate, branched, smooth, thin-walled, 2–2.5 µm wide. Conidiomata pycnidial, dark brown, semi-immersed, solitary, scattered, globose to subglobose 75–120 × 70–130 µm, conidiomata wall 4–6-layered, 10–20 µm thick, with an outer layer of *textura angularis*, composed of brown to dark brown, flattened polygonal cells of 4–6 µm diameter. Conidiophores absent. Conidiogenous cells phialidic, determinate, hyaline, smooth, thin-walled, flask-shaped, ampulliform, or cylindrical, 4–6 × 4–8 µm. Conidia aseptate, hyaline, smooth, thin-walled, bacilliform, globose to kidney-shaped, 2–3 × 1.5–2.0 µm. Chlamydospores absent.Culture characteristics: Colonies on PDA reaching 16–18 mm diameter after 7 days at 25 + 1 °C, flattened, velvety, margin regular, brownish grey to orange-grey (6C2/6B2); reverse brownish grey to orange-grey (6F2/6A2). Colonies on OA reaching 14–16 mm diameter after 7 days at 25 + 1 °C, flattened, velvety, margin regular, surface, and reverse brownish beige to grey (6F3/6B1). Colonies on MEA reaching 10–15 mm diameter after 7 days at 25 + 1 °C, flattened, velvety, margins undulate, surface, and reverse greyish brown to grey (6F3/6B1). Exopigment absent. Cardinal temperatures for growth: optimum 25 °C, maximum 30 °C, minimum 5 °C.*Diagnosis*: Morphologically, A. asexualis differs from the rest of the species of the genus because it is the only species that has a coelomycetous asexual state.*Notes*: Amniculicola asexualis is phylogenetically close to *A. gutulata*. The difference in nucleotides between both concatenated (ITS-tef1) sequences is 27 bp (Figure 3).

Key to freshwater Amniculicola species (modified from Dong et al. [10])1. Asexual state present.................................................................................................21. Sexual state present only...........................................................................................32. Conidia elongate to sigmoidal, produced from conidiophores........*A. longissima*2. Conidia bacilliform, globose to reniform, produced in pycnidia........*A. asexualis*3. Ascomata superficial.................................................................................................43. Ascomata immersed.................................................................................*A. immersa*4. Asci longer than 130 µm..........................................................................*A. lignicola*4. Asci shorter than 130 µm.........................................................................................55. Substrate stained purple...............................................................................*A. parva*5. Unstained substrate..................................................................................................66. Peridium 35–50 µm thick.........................................................................*A. aquatica*6. Peridium 27–35 µm thick........................................................................*A. guttulata*


Roussoellaceae J.K. Liu, Phookamsak, D.Q. Dai, and K.D. Hyde, Phytotaxa 181: 7 (2014) MycoBank MB 804651.Type genus: *Roussoella* Sacc., Atti dell’Istituto Veneto Scienze 6: 410 (1888) MycoBank MB 4799.*Elongatopedicellata* J.F. Zhang, J.K. Liu, K.D. Hyde, and Z.Y. Liu, Fungal Diversity 75: 118 (2015) MycoBank MB 551484Type species: *Elongatopedicellata lignicola* J.F. Zhang, J.K. Liu, K.D. Hyde, and Z.Y. Liu, Fungal Diversity 75: 118 (2015) MycoBank MB 551485.Because the asexual state of the genus *Elongatopedicellata* has not been reported and described until now, we have emended the generic description as follows:*Elongatopedicellata* J.F. Zhang, J.K. Liu, K.D. Hyde, and Z.Y. Liu, emended by V. Magaña-Dueñas, Cano, and Stchigel.


Hyphae hyaline to brown, septate, branched, smooth, thin-walled. Sexual state: Ascomata solitary to gregarious, scattered, immersed or erumpent, uniloculate, subglobose to obpyriform, coriaceous, with papillate ostiole; peridial wall 14–21 μm thick, composed of several layers of brown to dark brown, thick-walled cells of *textura angularis*. Hamathecium composed of 1–2 μm wide, filiform pseudoparaphyses, anastomosing between and above the asci, embedded in a gelatinous matrix. Asci 8-spored, bitunicate, fissitunicate, fusiform-clavate, with a long pedicel, apically rounded, with a well-developed ocular chamber. Ascospores 1–3 overlapping seriate, hyaline, fusiform, 1–septate, constricted at the septum, upper cell shorter and wider, lower cell long and narrow, surrounded by a mucilaginous sheath. Asexual state: Conidiomata pycnidial, semi-immersed, solitary, scattered, ostiolate, setose. Conidiomata wall of *textura intricata*, pale brown from the base to the middle part of the fruiting body, darkening towards the top, globose. Conidiophores absent. Conidiogenous cells phialidic, determinate, hyaline, smooth-walled, flask-shaped to ampulliform. Conidia aseptate, smooth-walled, hyaline to pale brown, clavate, ovoid, or kidney-shaped.
*Elongatopedicellata aquatica* V. Magaña-Dueñas, Cano, and Stchigel, sp. nov. MycoBank MB 551484. (Figure 10).Etymology: From Latin *aquaticus*, referring to the habitat from which the fungus was isolated.Type: Spain, Tarragona province, Capafonts, *de les Hortes river*, from plant debris submerged in freshwater, Mar 2019, Viridiana Magaña Dueñas and Isabel Iturrieta González, holotype CBS H-24926, living cultures FMR 17834, CBS 148920.

Description: Sexual state: unknown. Asexual state (found in natural substrate but described in vitro): Hyphae hyaline to light brown, septate, branched, smooth, thin-walled, 1.0–2.5 µm wide. Conidiomata pycnidial, semi-immersed, solitary, scattered, ostiolate, setose, subglobose, 160–190 × 165–220 µm, ostiole 15–20 µm diameter, setae hyaline to brown, septate, erect, nodose, curved at the tip, sometimes narrowing towards the tip, 60–90 µm long, mainly disposed around the ostiole, conidiomata wall 6–8-layered, 15–25 µm thick, with an outer layer of *textura intricata* composed of hyaline to dark brown hyphae 1.5–2.5 µm diameter, and an inner wall composed by hyaline flattened cells, the basal part of the pycnidium is pale brown towards the middle part, then darker towards the tip. Conidiophores absent. Conidiogenous cells phialidic, determinate, hyaline, smooth, thin-walled, ampulliform to globose, 5–6 × 4–5 µm. Conidia aseptate, smooth, thin-walled, hyaline to pale brown, brown in mass, clavate, ovoid or reniform, 3.5–5.5 × 1.5–3.5 µm. Chlamydospores absent.

Culture characteristics: Colonies on PDA reaching 19–20 mm diameter after 7 days at 25 + 1 °C, convex, cotton-like, margin regular, brownish grey to orange-grey (6D2/6B2), reverse orange-white to orange-grey (5A2/6B2). Colonies on OA reaching 20–22 mm diam after 7 days at 25 + 1 °C, flattened, matte, margin regular, surface and reverse orange-white (6A2). Colonies on MEA reaching 18–19 mm diameter after 7 days at 25 + 1 °C, flattened, matte, margin regular, surface and reverse yellowish white (3A2). Exopigment absent. Cardinal temperatures for growth: optimum 25 °C, maximum 35 °C, minimum 5 °C.

Diagnosis: Up to the present work, the genus Elongatopedicellata was considered to be monospecific. Elongatopedicellata lignicola, its type species was described as producing an asexual state only on wood [37]. In our study, we report a new species, *E. aquatica*, characterized by the production of an asexual coelomycetous state on the natural substrate as well as in vitro.

Notes: The difference in nucleotides between LSU sequences of *E. lignicola* and *E. aquatica* is 27 bp.
Lophiostomataceae Sacc, Sylloge Fungorum 2: 672 (1883) MycoBank MB 80966.Type genus: *Lophiostoma* Ces. and De Not., Commentario della Società Crittogamologica Italiana 1: 219 (1863). MB 2933.*Neovaginatispora* A. Hashim., K. Hiray., and Kaz. Tanaka, Studies in Mycology 90: 167 (2018). MycoBank MB 823147.Type species: *Neovaginatispora fuckelii* (Sacc.) A. Hashim., K. Hiray., and Kaz. Tanaka, Studies in Mycology 90: 167 (2018). MycoBank MB 823148.≡ *Lophiostoma fuckelii* Sacc., Michelia 1 (no. 3): 336 (1878).=*Vaginatispora fuckelii* (Sacc.) Thambug et al., Fungal Diversity 74: 243. (2015).

Because the asexual state of *Neovaginatispora* has not been reported and described up to this paper, we have emended the description of this genus as follows:*Neovaginatispora* A. Hashim., emended by V. Magaña-Dueñas, Cano, and Stchigel.

*Description*: Hyphae pale brown, branched, septate, smooth, thin-walled. Sexual state: Ascomata solitary, semi-immersed to erumpent, black, coriaceous, ostiolate, subglobose. Ostiole rounded or slit-like, central, with a pore-like opening. Peridium uneven in width, thinner at the base, two-layered, outer layer fusing with the host cells, inner layer comprising hyaline cells of *textura angularis*. Hamathecium composed of numerous, cellular, hypha-like, septate pseudoparaphyses. Asci 8-spored, bitunicate, fissitunicate, cylindric-clavate, with a short, bulbous pedicel, with an indistinct ocular chamber; ascospores uni- to biseriate, two- to multi-celled, constricted at the septa, hyaline, smooth, thin-walled, fusiform with acute ends, with globose appendages at both ends. Asexual state: Conidiomata pycnidial, brown, towards the ostiole dark brown, semi-immersed, solitary, scattered, globose, ostiolate, conidiomata wall of *textura angularis*, composed of brown to dark brown polygonal cells. Conidiophores absent. Conidiogenous cells phialidic, determinate, hyaline, smooth, thin-walled, globose. Conidia aseptate, hyaline, smooth- and thin-walled, guttulate. Chlamydospores abundant, aseptate, terminal, and intercalary, sometimes in chains, thick-walled, brown, globose to clavate.
*Neovaginatispora aquadulcis* V. Magaña-Dueñas, Cano, and Stchigel, sp. nov. MycoBank MB842771 (Figure 11).

Etymology. From Latin *aqua*-, water; and -*dulcis*, fresh, because of the habitat of the fungus.

Type: Spain, Castellón province, Burriana, Clot de la Mare de Déu, from plant debris submerged into freshwater, Mar 2021, Alan Omar Granados Casas and Ana Fernández Bravo, holotype CBS H- 24927, living cultures FMR 18914, CBS 148921.

Description: The fungus produces the sexual state on natural substrate. In vitro, both states are produced. Sexual state (in vitro): Hyphae septate, pale brown, branched, smooth, thin-walled, 2–3 µm wide. Sexual state: Ascomata perithecial, immersed to semi-immersed, solitary, dark brown, papillate, glabrous, pyriform, 400–550 × 300–325 µm, neck conic-truncate, 100–125 × 90–100 µm; peridial wall 3–4-layered, 40–60 µm thick, outer wall of *textura angularis* composed of dark brown flattened polygonal cells 6–8 µm diameter. Hamathecium comprising numerous hyaline, filamentous, septate, branched paraphyses at the base, 1–1.5 µm wide. Asci 8-spored, bitunicate, cylindrical to cylindrical-clavate, 50–70 × 8–10 µm, without apical structures. Ascospores 3-septate, hyaline, fusiform, 15–16 × 4–4.5 µm, with papillate to pulvinate appendages at each end. Asexual state (in vitro): Conidiomata pycnidial, brown, towards the ostiole dark brown, semi-immersed, solitary, scattered, ostiolate, globose, 165–180 × 160–170 µm, ostiole 20–30 µm diameter, conidiomata wall 4–6-layered, 15–20 µm thick, with an outer layer of *textura angularis* composed of brown to dark brown polygonal cells 5–10 µm diameter. Conidiophores absent. Conidiogenous cells phialidic, determinate, hyaline, smooth- and thin-walled, globose, 5–6 × 4–5 µm. Conidia aseptate, hyaline, smooth, thin-walled, guttulate, 2.5–3.5 × 1.5–2.5 µm. Chlamydospores abundant, aseptate, terminal, and intercalary, sometimes in chains, thick-walled, brown, globose to clavate 6–8 × 5–6 µm.

Culture characteristics: Colonies on PDA reaching 8–10 mm diameter after 7 days at 25 + 1 °C, convex, velvety, margin regular, grey (30E1), reverse orange-grey (6B2). Colonies on OA reaching 6–9 mm diameter after 7 days at 25 + 1 °C, convex, velvety, margin regular, surface, and reverse brownish grey (5F2). Colonies on MEA reaching 7–10 mm diameter after 7 days at 25 + 1 °C, convex, velvety, border regular, surface, and reverse grey to orange-grey (6D1/6B2). Exopigment absent. Cardinal temperatures for growth: Optimum 25 °C, maximum 35 °C, minimum 5 °C.

Diagnosis: *Neovaginatispora aquadulcis* is the only species of the genus that produces both asexual and sexual morphs in vitro. Furthermore, our species is characterized by the production of ascospores 1–3-septate, unlike the two previously reported species (1-septate).

Notes: Differences in nucleotides between *N. aquadulcis* and *N. fuckelii*, and *N. clematidis* ITS-*rpb2-tef1* concatenated sequences are 47 and 25 bp, respectively.
Chaetomellaceae Baral, P.R. Johnst., and Rossman, Index Fungorum 225: 1 (2015). MycoBank MB 551076

Type genus: *Chaetomella* Fuckel, Fungi Rhenani Exsiccati. Supplementi Fasc. 5: no. 1962 (1867). MycoBank MB 7575.

*Pilidium* Kunze in Kunze and Schmidt, Mykol. Hefte 2: 92 (1823). MycoBank MB 9395.
= *Sclerotiopsis* Speg., An. Soc. Cient. Argent. 113: 14 (1882).= *Hainesia* Ellis and Sacc., in Saccardo, Syll. fung. (Abellini) 3: 698 (1884).= *Discohainesia* Nannf., Nova Acta Regiae Soc. Sci. Upsal., Ser. 48: 88 (1932).

Type species: *Pilidium acerinum* Kunze in Kunze and Schmidt, Mykol. Hefte 2: 92 (1823). MycoBank MB 178919.
*Pilidium cuprescens* V. Magaña-Dueñas, Cano, and Stchigel, sp. nov. MycoBank MB842772 (Figure 12).

Etymology. Referring to the copper-colored conidiomata.

Type: Spain, Sevilla province, *Parque Natural Sierra Norte, Cascadas del Huéznar*, from plant debris in freshwater, May 2019, José F. Cano Lira and Juan R. García Martínez, holotype CBS H-24928, living cultures FMR 17839, CBS 148922.

Description: Sexual state: unknown. Asexual state (produced on natural substrate and in vitro; description in vitro): Hyphae hyaline to subhyaline, septate, branched, smooth, thin-walled, 2–2.5 µm wide. Conidiomata pycnidial, copper-colored, semi-immersed, solitary, scattered, non-ostiolate, globose to subglobose 300–360 × 280–350 µm, conidiomata wall 4–6-layered, 10–20 µm thick, with an outer layer of *textura angularis*, composed of copper-colored, flattened polygonal cells of 4–6 µm diameter. Conidiophores hyaline, septate, straight or sinuous to slightly curved, some branching at the base, 15–30 × 1–2 µm. Conidiogenous cells phialidic, integrated into the conidiophore, terminal and lateral, hyaline, smooth, thin-walled. Conidia aseptate, hyaline, smooth, thin-walled, fusiform, 10–15 × 2–3 µm. Chlamydospores absent.

Culture characteristics: Colonies on PDA reaching 23–25 mm diameter after 7 days at 25 + 1 °C, convex, concentrically radiate, margin lobed, greyish orange to brownish orange with a ring reddish brown (6B3/6C4/8F7); reverse brown to orange-grey (6E6/6B2). Colonies on OA reaching 8–10 mm diameter after 7 days at 25 + 1 °C, flattened, with abundant production of pycnidia that provide a granular texture, margin regular, surface, and reverse orange-white (6A2). Colonies on MEA reaching 12–13 mm diameter after 7 days at 25 + 1 °C, flattened, matte, margin regular, surface and reverse greyish yellow to yellowish white (4B4/4A2). Exopigment absent. Cardinal temperatures for growth: optimum 25 °C, maximum 35 °C, minimum 5 °C.

Diagnosis: Species of *Pilidium* are characterized by producing two sorts of conidiomata, pycnidial (pale brown when young, dark brown to black at maturity and uniloculate), and sporodochial (stalked, pale brown near base, becoming dark brown at apex, or reddish brown) [38,39,40,41,42]. As well as all the other species of the genus, *P. cuprescens* produces the typical hyaline, septate, branching conidiophores bearing acropleurogenous phialides, and hyaline, non-septate, fusiform, curved conidia. However, *P. cuprescens* is easily distinguished by the production of closed copper-colored pycnidia, while the sporodochial conidiomata have never been seen in culture (reported for *P. lythri* and *P. pseudoconcavum*).
Sympoventuriaceae Y. Zhang ter et al. Fungal Diversity 51: 255. (2011). MycoBank MB 563117.Type genus: Sympoventuria Crous and Seifert, Fungal Diversity 25: 31. 2007. MycoBank MB 501002.Pseudosigmoidea K. Ando and N. Nakam., J. Gen. Appl. Microbiol., Tokyo 46: 55. 2000. MycoBank MB 28418.Type species: Pseudosigmoidea cranei K. Ando and N. Nakam., J. Gen. Appl. Microbiol., Tokyo 46: 55. (2000). MycoBank MB 464825.*Pseudosigmoidea robusta* V. Magaña-Dueñas, Cano, and Stchigel, sp. nov. MycoBank MB842773 (Figure 13).

Etymology. From Latin *robustum*, robust, due to the nature of the conidiophores and the conidia.

Type: Spain, Segovia province, Riaza river, from plant debris submerged in freshwater, May 2018, Viridiana Magaña Dueñas, holotype CBS H-24929, culture ex type FMR 17416.

Description: Sexual state: unknown. Asexual stage (in vitro): Hyphae hyaline to subhyaline, septate, branched, smooth, thin-walled, 2.0–2.5 µm wide. Conidiophores pale brown, macronematous, mononematous, erect, straight or curved, geniculate, cylindrical, rounded at the tip, septate, 8–15 × 2–3 µm. Conidiogenous cells polyblastic, developing sympodially, terminal and intercalary, integrated to the conidiophore, 8–15 × 3–3.5 µm, denticulate. Conidia holoblastic, 1–9-septate, mostly solitary, subhyaline to pale brown, smooth, thin-walled, ovoid to obclavate, 8–65 × 2.5–3 µm, bearing a protruding foot-like scar due to a rhexolytic release from the conidiophore; in addition, the apical part of elongate conidia can develop a new conidiogenous locus, producing similar sort of conidia.

Culture characteristics: Colonies on PDA reaching 8–9 mm diameter after 7 days at 25 + 1 °C, umbonate, velvety, margin regular, surface and reverse greyish brown (7F3). Colonies on OA reaching 7–9 mm diameter after 7 days at 25 + 1 °C, flattened, velvety, margin regular, surface and reverse dark brown (7F3). Colonies on MEA reaching 5–7 mm diameter after 7 days at 25 + 1 °C, flattened, floccose, margin regular, surface and reverse greyish brown (7F3). Exopigment absent. Cardinal temperatures for growth: Optimum 20 °C, maximum 30 °C, minimum 5 °C.

Diagnosis: *Pseudosigmoidea robusta* is easily morphologically distinguishable from the other species of the genus because it produces well-developed conidiophores (semi-micronematous in the other species). The conidiophores also have widely separated conidiogenous loci and produce conidia very variable in shape. Furthermore, with the exception of *P. excentrica,* which forms shorter conidia (12–22 × 3–5 μm), the previously accepted species of Pseudosigmoidea produce larger conidia (80–250 × 3–4 μm in P. alnicola; 26–116.5 × 1.5–2.5 μm in P. cranei; and 68–133 × 4–8 μm in *P. ibarakiensis*) than *P. robusta* (8–65 × 2.5–3 μm).

Notes: Differences between the ITS-LSU nucleotide sequences of *P. robusta* and the other species of the genus are: from *P. alnicola*, 28 bp; from *P. excentricum*, 25 bp; and from *P. ibarakiensis*, 27 bp.
*Pyrenochaetopsidaceae* Valenzuela-Lopez, Cano, Guarro, Sutton, Wiederhold, Crous, and Stchigel, Stud. Mycol. 90: 56 (2017). MycoBank MB 820308.

Because the sexual state of the family has not been reported and described until now, we have emended the description of this family as follows:

Sexual state: Ascomata immersed to semi-immersed, brown to dark brown, ostiolate, outer wall of *textura angularis*. Hamathecium comprising hyaline, septate, filamentous paraphyses. Asci 8-spored, bitunicate, stipitate, cylindrical to clavate. Ascospores 3–6 septate, hyaline, fusiform. Asexual state: Conidiomata pycnidial, pale brown to brown, solitary or confluent. Conidiomata wall of *textura angularis*, glabrous or setose, subglobose to ovoid, with a non-papillate or papillate ostiolar neck. Conidiogenous cells phialidic, hyaline, discrete or integrated into the septate, acropleurogenous conidiophores. Conidia aseptate, hyaline, smooth, thin-walled, ovoid, cylindrical to allantoid, guttulate.
Type genus: *Pyrenochaetopsis* Gruyter, Aveskamp, and Verkley, Mycologia 102: 1076 (2010). MycoBank MB 514653.

Since the sexual state of the genus *Pyrenochaetopsis* has not been reported before, below we emended the generic description.

Sexual state: Ascomata immersed to semi-immersed, brown to dark brown, ostiolate, outer wall of *textura angularis.* Hamathecium comprising hyaline, septate, filamentous paraphyses. Asci 8-spored, bitunicate, stipitate, cylindrical to clavate. Ascospores 3–6 septate hyaline, fusiform, mostly biseriate within the ascus. Asexual state: Conidiomata pycnidial, honey to citrine or olivaceous to olivaceous black, solitary to confluent, superficial or submerged, with a non-papillate or papillate ostiolar neck. Conidiomata wall pseudoparenchymatous, setose, globose to subglobose. Conidiogenous cells phialidic, hyaline, discrete, and integrated into acropleurogenous conidiophores. Conidia aseptate, cylindrical to allantoid, guttulate.
Type species: *Pyrenochaetopsis leptospora* (Sacc. and Briard) Gruyter, Aveskamp, and Verkley, Mycologia 102: 1076 (2010). MycoBank MB 514654.*Pyrenochaetopsis perfecta* V. Magaña-Dueñas, Stchigel, and Cano, sp. nov. MycoBank MB842774 (Figure 14).


Etymology: From Latin *perfectus*, perfect, because the fungus produces both sexual and asexual states.Type: Spain, Castellón province, Burriana, *Clot de la Mare de Déu*, from freshwater submerged plant debris, Mar 2021, Alan Omar Granados Casas and Ana Fernández Bravo, holotype CBS H-24930, living cultures FMR 18913, CBS 148923.


Description: Sexual state (on natural substrate): Ascomata immersed to semi-immersed, brown to dark brown, ostiolate, outer wall of *textura angularis*. Hamathecium comprising hyaline, septate, filamentous paraphyses, 1–3 µm wide. Asci 8-spored, bitunicate, stipitate, cylindrical to clavate, 70–100 × 8–10 µm, stipe 10–15 µm long. Ascospores 3–6 septate hyaline, fusiform, 25–26 × 4–5 µm. Asexual state (in vitro): Conidiomata pycnidial, semi-immersed, light brown to brown with dark brown patches, solitary, scattered, setose, subglobose, 240–380 × 260–300 µm, setae pale brown to brown, septate, sinuous, thick-walled, 30–150 µm, rounded and curved at the tip, conidiomata wall 4–6-layered, 15–20 µm thick, outer layer of *textura angularis* composed of light brown to dark brown, flattened polygonal cells of 8–10 µm diameter, wrapped in a framework of nodose, branching, sinuous and thick-walled hyphae. Conidiogenous cells phialidic, determinate, hyaline, ampulliform, smooth-walled, 3.5–4 × 2.5–3.5 µm. Conidia aseptate, hyaline, smooth, thin-walled, ellipsoidal, 4.5–5 × 1.5–2 µm. Chlamydospores absent.

Culture characteristics: Colonies on PDA reaching 33–35 mm diameter after 7 days at 25 + 1 °C, flattened, velvety, margin regular, surface and reverse grey (6D1) margin orange-white (6A2). Colonies on OA reaching 33–35 mm diameter after 7 days at 25 + 1 °C, flattened, velvety, margin regular, surface, and reverse grey (5B1). Colonies on MEA reaching 28–30 mm diameter after 7 days at 25 + 1 °C, flattened, velvety, margins undulate, grey to orange-grey (5C1/5B2); reverse grey to yellowish brown (5F1/5E4). Exopigment absent. Cardinal temperatures for growing: optimum 25 °C, maximum 35 °C, minimum 5 °C.

Diagnosis: Similarly to the other species of the genus, *P. perfecta* produces pycnidia ornamented with brown setae (especially abundant around and near the ostiole) and an abundant amount of hyaline, one-celled small phialoconidia. *Pyrenochaetopsis perfecta* is easily distinguishable from the other species of the genus because is the only one that produces a sexual state. Unlike its closest species, *P. globosa, P. perfecta* produces bigger pycnidia (240–380 × 260–300 µm vs. 50–220 × 140–190 μm) covered by abundant long setae (absent in *P. globosa*).
Pyrenochaetopsis cylindrospora V. Magaña-Dueñas, Stchigel, and Cano, sp. nov. MycoBank MB842778 (Figure 15).


Etymology: From Greek κυλινδρικός-, cylindrical, and -σπόριο, spore, because of the shape of the conidia.Type: Spain, Castellón province, Burriana, Clot de la Mare de Déu, from freshwater submerged plant debris, Mar 2021, Alan Omar Granados Casas and Ana Fernández Bravo, holotype CBS H-24931, living cultures FMR 18801, CBS 148924.


Description: Sexual state: unknown. Asexual state (produced on natural substrate and in vitro; description in vitro): Hyphae hyaline to pale brown, septate, branched, smooth, thin-walled, 2–3 µm wide. Conidiomata pycnidial, brown to dark brown, immersed to semi-immersed, solitary, scattered, setose, ostiolate, subglobose, 100–120 × 110–130 µm, ostiole 10–20 µm diameter, setae brown, septate, slightly warty, sinuous, rounded at the tip, 35–75 µm long 2–2,5 µm wide, conidiomata wall 4–6–layered, 15–25 µm thick, with an outer layer of *textura angularis*, composed of pale brown to dark, flattened polygonal cells of 4–8 µm diameter. Conidiophores absent. Conidiogenous cells phialidic, determinate, hyaline, smooth, thin-walled, cylindrical, 4–5 × 2–3 µm. Conidia aseptate, hyaline, smooth, thin-walled, guttulate, bacilliform, 5–6 × 1.5–2 µm. Chlamydospores absent.

Culture characteristics: Colonies on PDA reaching 15 mm diameter after 7 days at 25 ± 1 °C, umbonate, slightly velvety, margin regular, grey to orange-grey (6C1/6C2), border white; reverse brownish grey to orange-grey (6D2/6C2). Colonies on OA reaching 12 mm diameter after 7 days at 25 ± 1 °C, flattened, slightly floccose, margin regular, surface, and reverse grey to orange-grey (5F1/6B2). Colonies on MEA reaching 13 mm diameter after 7 days at 25 ± 1 °C, flattened, velvety, margins regular, brown to orange-grey surface and reverse (5F6/5B2). Exopigment absent. Cardinal temperatures for growing: optimum 25 °C, maximum 30 °C, minimum 5 °C.
Diagnosis: The closest species to *P. cylindrospora* is *P. confluens*. Morphologically, *P. cylindrospora* differs from *P. confluens* because the former produces lager setae (35–75 µm vs. 5–22.5 (–35) µm) and bigger conidia (5–6 × 1.5–2 µm vs. 2–4 × 2–2.5 μm).Notes: The nucleotide difference between *P. cylindrospora* and *P. confluens* ITS-rpb2-tub2 concatenated sequences is 24 bp.

## 4. Discussion

In this work, we report the finding of seven new species of presumptively facultative freshwater ascomycetes belonging to the families Amniculicolaceae *(Amniculicola asexualis), Chaetomellaceae (Pilidium cuprescens), Lophiostomataceae (Neovaginatispora aquadulcis), Pyrenochaetopsidaceae (Pyrenochaetopsis cylindrospora* and *P. perfecta*), Roussoellaceae (Elongatopenicillata aquatica), and *Sympoventuriaceae (Pseudosigmoidea robusta*), all of which were isolated from decomposing submerged plant material in lotic freshwater environments.

The genus *Amniculicola* was introduced by Zhang et al. [43] to accommodate *A. lignicola*. Until now, six species were accepted in the genus (http://www.indexfungorum.org/names/Names.asp; accessed on 3 April 2022), and all of them were discovered in freshwater. Most of these species stain the substrate in purple tinges [43,44], with the exceptions of *A. aquatica, A. guttulata* [10,44], and the new species, *A. asexualis. Amniculicola longissima* was the only species previously reported as producing an asexual state, characterized by the formation of long curved or sigmoid conidia on short coniodiophores of sympodial development [45]. *Amniculicola asexualis* is the only species producing a coelomycetous asexual state and lacking a sexual state.

*Elongatopedicellata* was introduced by Zhang et al. [37] to accommodate *E. lignicola*. This genus only comprises the type species, isolated from a dead tree branch in Mae Chang Hot Spring, Thailand. *Elongatopedicellata lignicola* produces papillate ascomata, filiform pseudoparaphyses, bitunicate, fissitunicate, and fusiform-clavate asci, and hyaline, 1-septate, fusiform ascospores constricted at the septum. The authors did not report the production of any fertile structure in pure culture [37]. The new species, *E. aquatica*, is the only one from aquatic habitats that is characterized by the production of pyrenochaeta-like or pyrenochaetopsis-like conidiomata, an undescribed feature for the genus.

Hashimoto et al. [46] performed a phylogenetic study of the *Lophiostomataceae* and introduced the genus *Neovaginatispora* to accommodate *N. fuckelii*. This genus differs from *Vaginatispora* in having thinner, sub-carbonaceous peridium of uniform thickness [45,46]. Currently, *Neovaginatispora* comprises two species, *N. clematidis* and *N. fuckelii*, both isolated from decaying plants (*Clematis viticella* and oak tree, and *Mangifera indica*, respectively) [46,47,48]. *Neovaginatispora fuckelii* has also been reported from wood submerged in freshwater [49]. The asexual state in both species is unknown. *Neovaginatispora aquadulcis*, recovered from plant debris in freshwater produce both sexual and asexual states in vitro. The coelomycetous asexual state in *N. aquadulcis* is a novel feature for that genus.

*Pilidium* is a genus introduced by Kunze and Schmidt [50]. Species of Pilidium are commonly found as plant-associated fungi or isolated from soil, and they are known to produce two kinds of conidiomata: sporodochia and pycnidia [51]. Until recent years, the genus included *P. acerinum, P. eucalyptorum, P. lythri, P. pseudoconcavum,* and *P. septatum* [38,39,42]. Subsequently, Crous et al. [40,41] introduced P. anglicum and *P. novae-zelandie*. From all known species, only *P. septatum* has been reported in freshwater environments [42]. The new species, *Pilidium cuprescens*, also found in freshwater, is easily distinguished from other species of the genus by the production of closed, copper-colored globose conidiomata.

The genus *Pseudosigmoidea* was created by Ando and Nakamura [52] to place the fungal strain 85B-65 (= ATCC 16660) isolated from freshwater in Maryland (USA). Originally, this fungus was identified as Sigmoidea (≡ Flagellospora) prolifera. Later, Ando and Nakamura [52] distinguished *P. cranei* (the type species of the new genus) from *S. prolifera* because the former produces enteroblastic conidia from polyphialides, and the second one holoblastic conidia on a sympodialy proliferative conidiogenous cell. Jones et al. [53], in a phylogenetic study based on the analysis of the nucleotide sequences of the small subunit (SSU) of the ribosomal DNA, confirmed *P. cranei* as a distinct taxon from *S. prolifera*, despite both fungi being placed in the family Phaeosphaeriaceae. Later, Diene et al. [54] described the second species, *P. ibarakiensis*, from forest soil in Honshu (Japan). Despite the SSU sequences displaying a high similarity (99 %) with those of Troposporella fumosa, Helicoma monilipes, and H. olivaceum, the fungus was placed as a new species of Pseudosigmoidea because it showed a high alignment score (100%) and 97 % similarity with the SSU sequence of *P. cranei*, but also based on morphological similarities (*P. cranei* produce long subcylindrical to obclavate conidia and *P. ibarakiensis* scolecoid conidia, whereas these spores are helical in *T. fumosa, H. monilipes,* and *H. olivaceum*). However, the authors [54] did not build a phylogenetic tree to show the relationships between these taxa. *Pseudosigmoidea alnicola*, isolated from leaf litter near Berlin (Germany), was the third species of the genus. Despite a phylogenetic tree being built, only the ITS and LSU nucleotide sequences of the new species were used, and consequently, *P. alnicola* falls into the same terminal clade as *T. fumosa*, *Scolecobasidium excentricum*, *Sympoventuria capensis,* and *Sympoventuria melaleucae*. However, the authors mentioned that the ITS sequence of *P. alnicola* displays 95 % similarity with that of *P. ibarakiensis* [55]. The morphological similarity between the reproductive structures of *P. alnicola, P. cranei,* and *P. ibarakiensis* was probably responsible for the final placement at genus level. The fourth species, *P. excentrica* [56], was originally erected as Scolecobasidium excentricum, being isolated from leaf litter in Santiago de las Vegas (Cuba) [57]. The morphology of *P. excentrica* differs from the other species of the genus because it produces larger conidiophores bearing sympodially proliferating conidiogenous cells (described as polyphialidic in the other species) and because it produces shorter, cylindrical to allantoid conidia with rounded ends (scolecoid in the remaining species). The morphology of our new species, *P. robusta*, is similar to *P. excentrica*, because it produces short cylindrical to obclavate conidia and conidiophores bearing integrated sympodially proliferating conidiogenous cells. However, *P. robusta* produces longer and narrower conidia than *P. excentrica* (8–65 × 2.5–3 µm vs. 14–22 × 3–5 µm), and in the apical part of the conidia, it can develop a conidiogenous locus (feature not reported for *P. excentrica*).

*Pyrenochaetopsis* was introduced by De Gruyter et al. [58] for several *phoma*-like species placed together in the same clade in a phylogenetic study. These authors introduced *P. decipiens, P. indica, P. leptospora, P. microspora,* and *P. pratorum* to that genus. Valenzuela-Lopez et al. [59], thanks to a multilocus phylogenetic analysis, transferred *Pyrenochaetopsis* from the family *Cucurbitariaceae* to the new family *Pyrenochaetopsidae*. The genus contains 19 species (http://www.indexfungorum.org/names/Names.asp; accessed on 3 April 2022). The members of this genus have been isolated from terrestrial, marine, and freshwater environments, and also from human clinical specimens (superficial tissue, bronchial washing, and blood) [16,58,59,60,61]. Up to now, only the asexual (coelomycetous) reproductive state has been described for the members of the genus and the family. In the present study, we describe two new species of *Pyrenochaetopsis. Pyrenochaetopsis perfecta* is the first one to produce a sexual reproductive state, characterized by the production of perithecial ascomata, cylindrical to clavate bitunicate asci and hyaline, 3–6-septate fusiform ascospores. In comparison with the second new species, *P. globosa, P. perfecta* produces bigger pycnidia covered by abundant long setae (structures absent in *P. globosa*).

## Figures and Tables

**Figure 1 jof-08-00849-f001:**
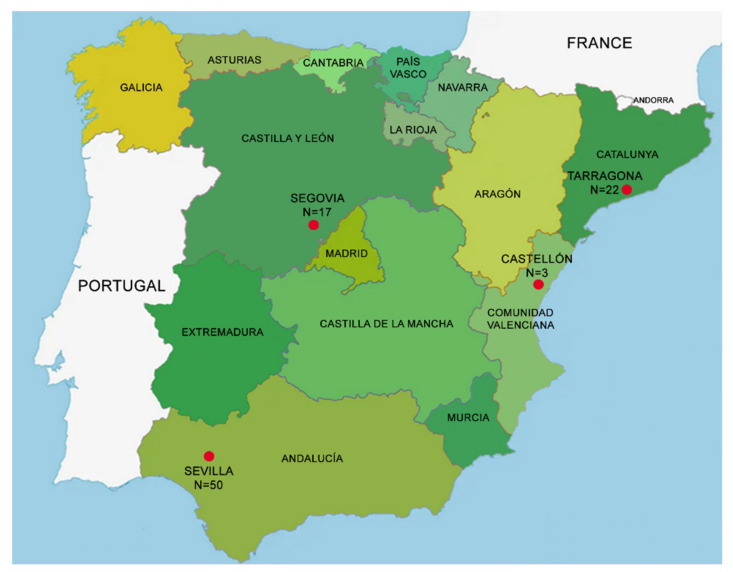
Sampling locations (https://www.huffingtonpost.es/entry/en-que-comunidad-autonoma-se-come-mejor_es_5d27582ee4b0bd7d1e18eed3; accessed on 3 April 2022).

**Figure 2 jof-08-00849-f002:**
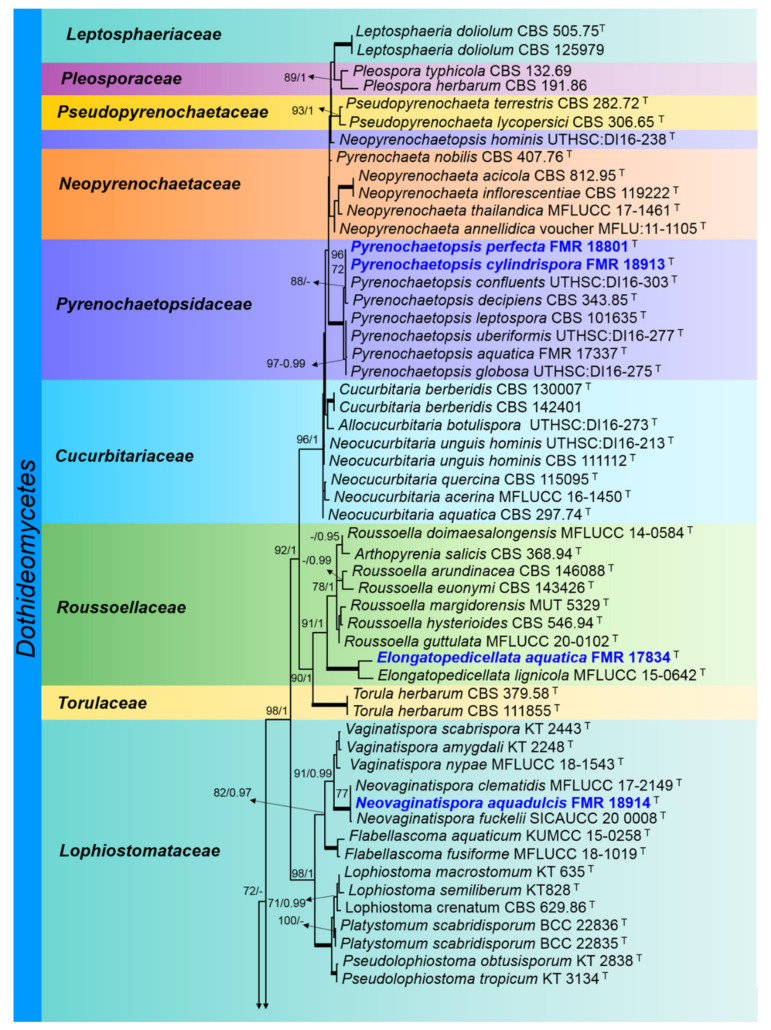
Phylogenetic tree inferred from an ML analysis of LSU nucleotide sequences of representatives of the *Dothideomycetes* and the *Leotiomycetes* classes. The RAxML bootstrap support values (BS) above 70% and the Bayesian posterior probabilities (PP) above 0.95 are given at the nodes (BS/PP). Fully supported branches (100 BS/1 PP) are indicated in thicker lines. Newly proposed taxa are given in **blue**. Type strains are indicated by a superscript “T”. The tree was rooted with *Saccharomyces cerevisiae* NRRL Y 12632 and M16. Alignment length 776 bp.

**Figure 3 jof-08-00849-f003:**
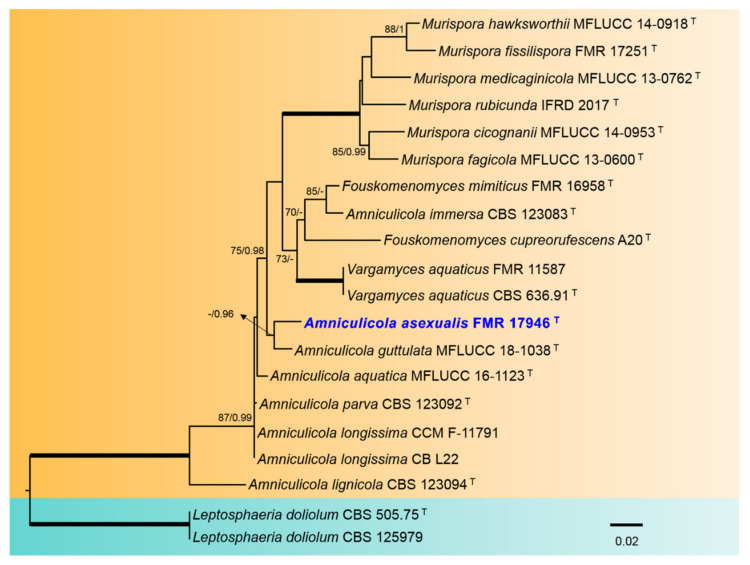
Phylogenetic tree inferred from an ML analysis based on a concatenated alignment of ITS and tef-1 sequences of 20 strains representing species in the *Amniculicolaceae*. The RAxML bootstrap support values (BS) above 70% and the Bayesian posterior probabilities (PP) above 0.95 are given at the nodes (BS/PP). Fully supported branches (100 BS/1 PP) are indicated in thicker lines. Newly proposed taxa are given in** blue**. Type strains are indicated by a superscript “T”. The tree was rooted with *Leptosphaeria dolium* CBS 505.75 and CBS 125979. Alignment length 1117 bp.

**Figure 4 jof-08-00849-f004:**
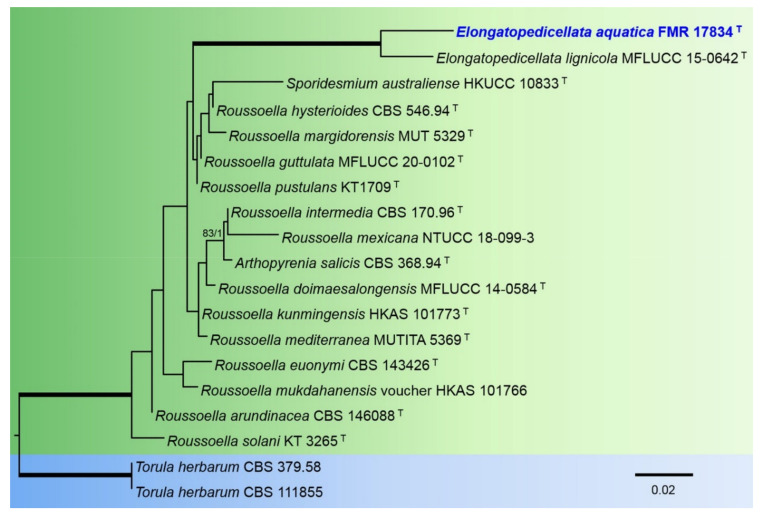
Phylogenetic tree inferred from an ML analysis inferred from LSU sequences of 18 strains representing species in the *Roussoellaceae*. The RAxML bootstrap support values (BS) above 70% and the Bayesian posterior probabilities (PP) above 0.95 are given at the nodes (BS/PP). Fully supported branches (100 BS/1 PP) are indicated in thicker lines. Newly proposed taxa are given in **blue**. Type strains are indicated by a superscript “T”. The tree was rooted with *Torula herbarum* CBS 379.58 and CBS 111855. Alignment length 845 bp.

**Figure 5 jof-08-00849-f005:**
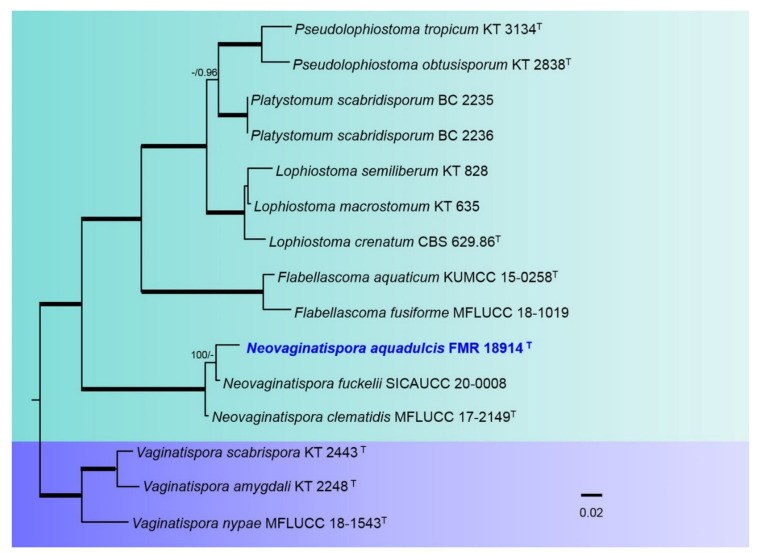
Phylogenetic tree inferred from an ML analysis based on a concatenated alignment of ITS, *rpb2,* and *tef-1* sequences of 15 strains representing species in the *Lophiostomataceae*. The RAxML bootstrap support values (BS) above 70% and the Bayesian posterior probabilities (PP) above 0.95 are given at the nodes (BS/PP). Fully supported branches (100 BS/1 PP) are indicated in thicker lines. Newly proposed taxa are given in **blue**. Type strains are indicated by a superscript “T”. The tree was rooted with *Vaginatispora amygdali* KT 2248, *V. nypae* MFLUCC 18-1543, and *V. scabrispora* KT 2443. Alignment length 2265 bp.

**Figure 6 jof-08-00849-f006:**
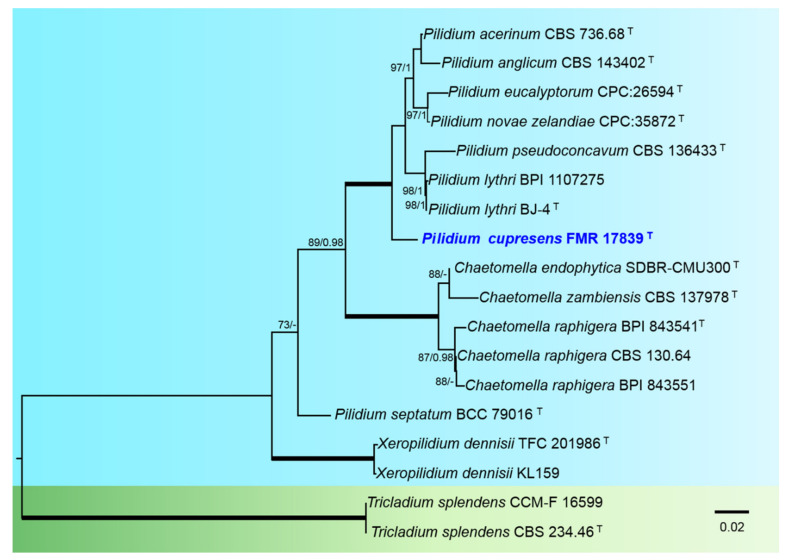
Phylogenetic tree inferred from an ML analysis based on a concatenated alignment of ITS, and LSU sequences of 18 strains representing species in the *Chaetomellaceae*. The RAxML bootstrap support values (BS) above 70% and the Bayesian posterior probabilities (PP) above 0.95 are given at the nodes (BS/PP). Fully supported branches (100 BS/1 PP) are indicated in thicker lines. Newly proposed taxa are given in **blue**. Type strains are indicated by a superscript “T”. The tree was rooted with *Tricladium splendens* CBS 234.46 and CCM-F 16599. Alignment length 1246 bp.

**Figure 7 jof-08-00849-f007:**
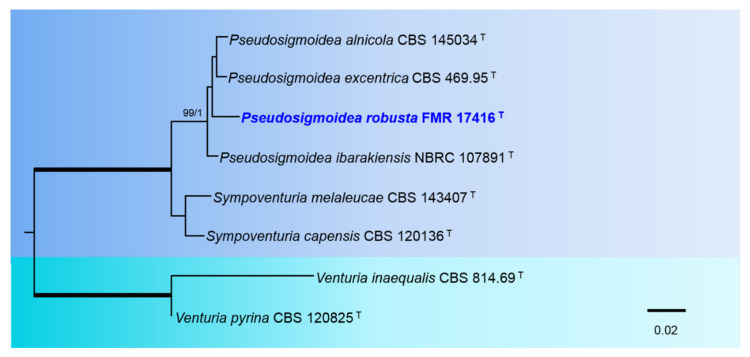
Phylogenetic tree inferred from an ML analysis based on a concatenated alignment of ITS, and LSU sequences of eight strains representing species in the *Sympoventuriaceae*. The RAxML bootstrap support values (BS) above 70% and the Bayesian posterior probabilities (PP) above 0.95 are given at the nodes (BS/PP). Fully supported branches (100 BS/1 PP) are indicated in thicker lines. Newly proposed taxa are given in **blue**. Type strains are indicated by a superscript “T”. The tree was rooted with *Venturia inaequalis* CBS 814.69 and *V. pyrina* CBS 120825. Alignment length 1325 bp.

**Figure 8 jof-08-00849-f008:**
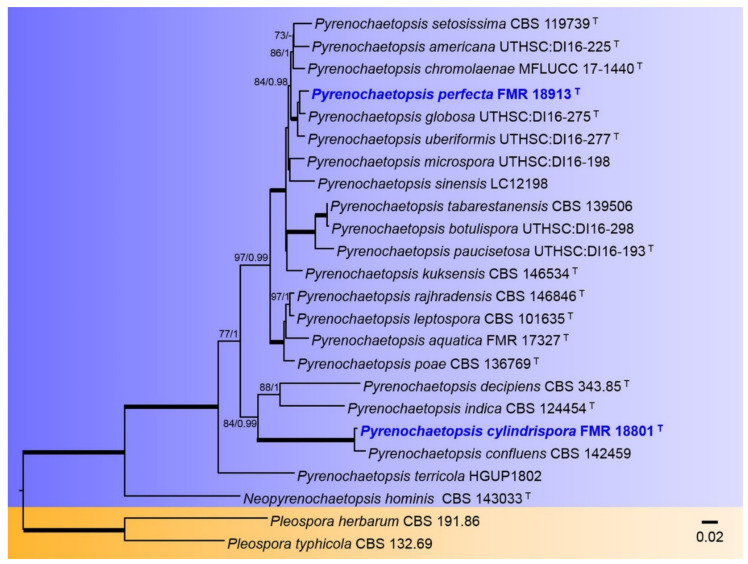
Phylogenetic tree inferred from an ML analysis based on a concatenated alignment of ITS, *rpb2,* and *tub2* sequences of 24 strains representing species in the *Pyrenochaetopsidaceae*. The RAxML bootstrap support values (BS) above 70% and the Bayesian posterior probabilities (PP) above 0.95 are given at the nodes (BS/PP). Fully supported branches (100 BS/1 PP) are indicated in thicker lines. Newly proposed taxa are given in **blue**. Type strains are indicated by a superscript “T”. The tree was rooted with *Pleospora herbarum* CBS 191.86 and *P. typhicola* CBS 132.69. Alignment length 1488 bp.

**Figure 9 jof-08-00849-f009:**
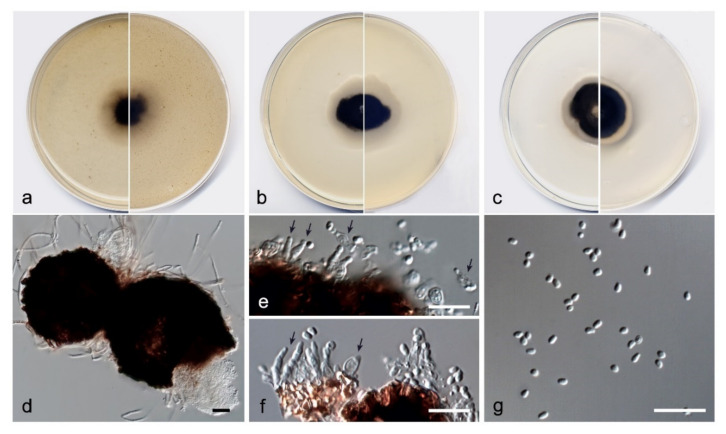
*Amniculicola asexualis* FMR 17946. (**a**) Colonies on OA (**b**) MEA (**c**) and PDA after two weeks at 25 + 1 °C (surface, left; reverse, right); (**d**) Pycnidia; (**e**,**f**) Conidiogenous cells (black arrows); (**g**) Conidia. Scale bars = 10 µm.

**Figure 10 jof-08-00849-f010:**
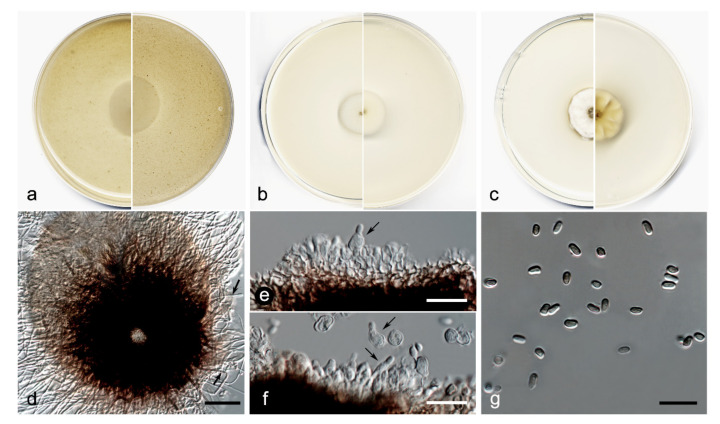
*Elongatopedicellata aquatica* FMR 17834 (**a**) Colonies on OA (**b**) MEA (**c**) and PDA after two weeks at 25 + 1 °C (surface, left; reverse, right); (**d**) Pycnidium (setae pointed out by black arrows); (**e**,**f**) Conidiogenous cells (black arrows); (**g**) Conidia. Scale bars: (**d**) = 50 µm, (**e**–**g**) = 10 µm.

**Figure 11 jof-08-00849-f011:**
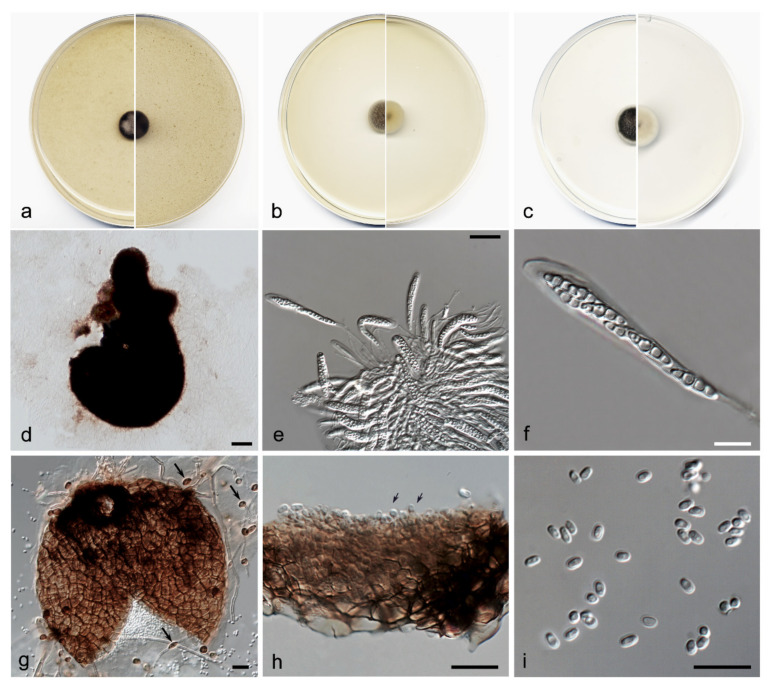
*Neovaginatispora aquadulcis* FMR 18914. (**a**) Colonies on OA (**b**) MEA (**c**) PDA after two weeks at 25 + 1°C (surface, left; reverse, right); (**d**) Ascomata; (**e**,**f**) Asci with ascospores; (**g**) Pycnidium and chlamydospores (black arrows); (**h**) Conidiogenous cells (black arrows); (**i**) Conidia. Scale bars: (**d**,**g**) = 50 µm, (**e**) = 25 µm, (**f**,**h**,**i**) = 10 µm.

**Figure 12 jof-08-00849-f012:**
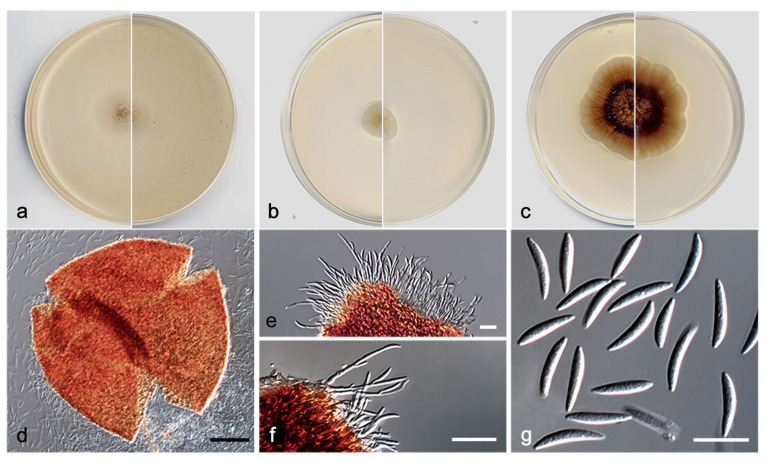
*Pilidium cuprescens* FMR 17839 (**a**) Colonies on OA (**b**) MEA (**c**) and PDA after two weeks at 25 ± 1 °C (surface, left; reverse, right); (**d**) Pycnidium; (**e**,**f**) Conidiophores; (**g**) Conidia. Scale bars: (**d**) = 50 µm, (**e**–**g**) = 10 µm.

**Figure 13 jof-08-00849-f013:**
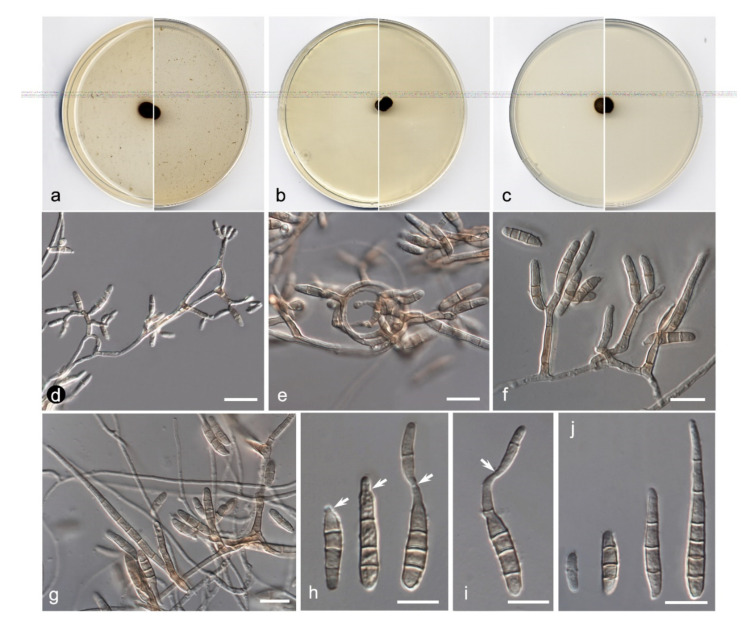
*Pseudosigmoidea robusta*. FMR (**a**) Colonies on OA (**b**) MEA (**c**) and PDA after two weeks at 25 ± 1 °C (surface, left; reverse, right); (**d**–**g**) Conidiophores; (**h**,**i**) Conidia displaying a secondary conidiogenesis (white arrows); (**j**) Conidia showing variation in size. Scale bars: (**d**–**j**) = 10 µm.

**Figure 14 jof-08-00849-f014:**
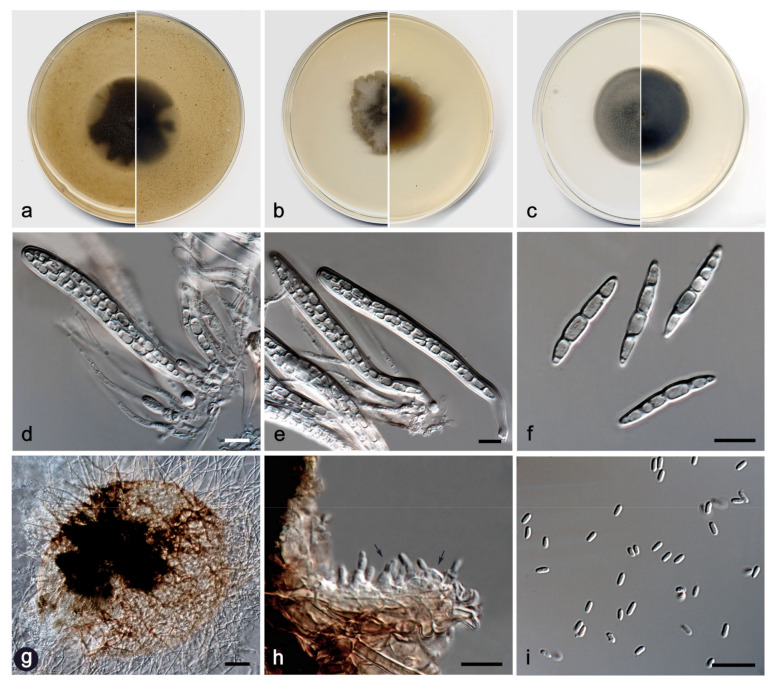
*Pyrenochaetopsis perfecta* FMR 18913 (**a**) Colonies on OA (**b**) MEA (**c**) and PDA after two weeks at 25 ± 1 °C (surface, left; reverse, right); (**d**,**e**) Asci with ascospores; (**f**) Ascospores; (**g**) Pycnidium; (**h**) Conidiogenous cells (black arrows); (**i**) Conidia. Scale bars: (**g**) = 25 µm, (**d**–**f**,**h**,**i**) = 10 µm.

**Figure 15 jof-08-00849-f015:**
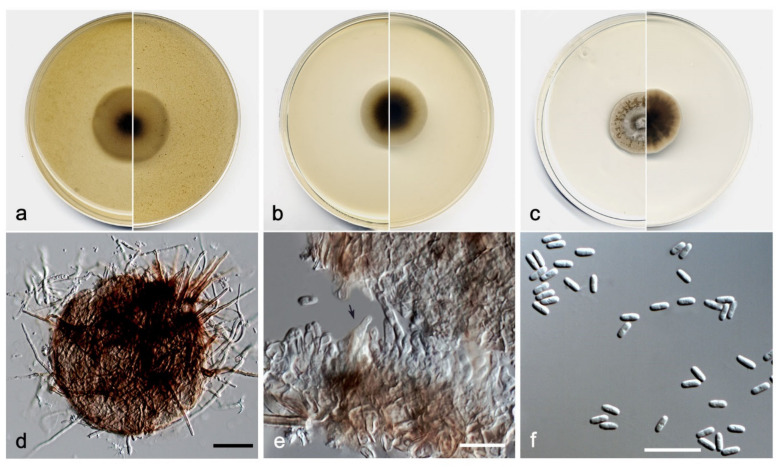
*Pyrenochaetopsis cylindrospora* FMR 18801 (**a**) Colonies on OA (**b**) MEA (**c**) and PDA after two weeks at 25±1 °C (surface, left; reverse, right); (**d**) Pycnidium; (**e**) Conidiogenous cells (black arrow); (**f**) Conidia. Scale bars: (**d**) = 20 µm, (**e**,**f**) = 10 µm.

## Data Availability

Not applicable.

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
