# Peer review of "Novel Freshwater Ascomycetes from Spain"

_jof, 2022, doi:10.3390/jof8080849_

Round 1

Reviewer 1 Report

I have made minor changes while reviewing the manuscript. These are self explanatory.

Author Response

Dear Reviewer 1,

All comments and changes suggested by reviewer 1 were taken into account and consequently implemented. Following, several clarifications and explanations in this regard are exposed.

Line 233-245: For clarity purposes move this section to line 226, after (90%BS/1 PP),

Response: We decided to leave the sentence as it originally was, since we consider it important to describe first the clade and then the distribution of the new taxa.

Line 318: has the type species P. cranei not been sequenced? Explain

Response: Only the SSU sequence of Pseudosigmoidea cranei is available on public databases, and such sequence was employed in a previous work to confirm its taxonomic placement as a separate genus from Sigmoidea (Gareth Jones, E.B, Zuccaro, A., Mitchell, J., Nakagiri, A., Chatmala, I., Pang, K-L. Phylogenetic position of freshwater and marine Sigmoidea species: introducing a marine hyphomycete Halosigmoidea gen. nov. (Halosphaeriales). Botanica Marina 52 (2009): 349–359. DOI 10.1515/BOT.2009.006). Because of the fact that the strain ATCC 16660 (formerly Sigmoidea prolifera (Petersen) Crane) is a very restricted item, it has not been sequenced again by other authors in later studies.

 Line 397: the name "microspora" is usually applied to a species when it is the one in the group with smaller spores compared to the rest. Maybe the name for this new species should reflect the fact that it is the only one having a coelomycetous asexual state.

Response: Changed by “asexualis”.

Line 502: a photo of free ascospores should be included in Fig. 11: showing the globose appendages at both ends

Response: We only observed a few ascospores showing inflated appendages on water mountings. Unfortunately, the holotype (preserved in lactophenol) only has ascospores inside the asci.

Line 527: generic description states that the appendages are globose!

Response: The morphology of the appendages has now been included in the manuscript. In the particular case of our species, they are papillate to pulvinate.

 Line 603: Could it be variation in maturity?

Response: Certainly, but we have observed this great variability also in mature conidia.

Sincerely,

The authors of the manuscript

Reviewer 2 Report

A manuscript describes seven new species of the Ascomycota found in various freshwater habitats in Spain. However, Amniculicola microspora is doubtful whether it is a new species in Amniculicola as details mentioned in the manuscript. Additional suggestions are also provided in the attached manuscript. In addition, an abstract should be rewritten. English editing is required. 

Author Response

Dear Reviewer,

All comments and changes suggested by reviewer were taken into account and consequently implemented. Attached, several clarifications and explanations in this regard are exposed.

Sincerely,

The authors of the manuscript

Round 2

Reviewer 2 Report

A minor comment from me. A full stop should be deleted from a title.

Please authors recheck again, and I have no more comments.